# Surgical Prehabilitation in Patients with Gastrointestinal Cancers: Impact of Unimodal and Multimodal Programs on Postoperative Outcomes and Prospects for New Therapeutic Strategies—A Systematic Review

**DOI:** 10.3390/cancers15061881

**Published:** 2023-03-21

**Authors:** Julie Mareschal, Alexandra Hemmer, Jonathan Douissard, Yves Marc Dupertuis, Tinh-Hai Collet, Thibaud Koessler, Christian Toso, Frédéric Ris, Laurence Genton

**Affiliations:** 1Clinical Nutrition, Service of Endocrinology, Diabetology, Nutrition and Therapeutic Education, Department of Medicine, Geneva University Hospitals, 1205 Geneva, Switzerland; alexandra.hemmer@hcuge.ch (A.H.);; 2Division of Abdominal Surgery, Department of Surgery, Geneva University Hospitals, 1205 Geneva, Switzerland; 3Department of Colorectal Surgery, Freeman Hospital—Newcastle upon Tyne Hospitals NHS Trust, Newcastle upon Tyne NE7 7DN, UK; 4Diabetes Centre, Faculty of Medicine, University of Geneva, 1205 Geneva, Switzerland; 5Department of Oncology, Geneva University Hospitals, 1205 Geneva, Switzerland

**Keywords:** oncology, digestive malignancy, preoperative intervention, nutrition, physical activity, probiotics, symbiotics, fecal microbiota transplantation, ghrelin receptor agonists

## Abstract

**Simple Summary:**

Gastrointestinal cancers comprise over 25% of new cancer cases. Surgery is the primary curative treatment. Prehabilitation before surgery aims to optimize the patient’s global condition to improve postoperative recovery. These programs usually include nutritional, physical activity, and/or psychological interventions. However, the benefits remain unclear. This review summarizes the latest evidence of preoperative prehabilitation on postoperative outcomes after gastrointestinal cancer surgery and discusses new potential therapeutic targets. Preoperative interventions, combining at least nutrition and physical activity, appear to improve physical performance, muscle strength, and quality of life in patients with esophagogastric and colorectal cancers. However, there was no benefit for postoperative complications, hospital length of stay, hospital readmissions, and mortality. Further studies are needed to confirm our findings, identify surgical cancer patients more likely to benefit from prehabilitation, harmonize interventions and integrate new therapeutic strategies.

**Abstract:**

The advantages of prehabilitation in surgical oncology are unclear. This systematic review aims to (1) evaluate the latest evidence of preoperative prehabilitation interventions on postoperative outcomes after gastrointestinal (GI) cancer surgery and (2) discuss new potential therapeutic targets as part of prehabilitation. Randomized controlled trials published between January 2017 and August 2022 were identified through Medline. The population of interest was oncological patients undergoing GI surgery. Trials were considered if they evaluated prehabilitation interventions (nutrition, physical activity, probiotics and symbiotics, fecal microbiota transplantation, and ghrelin receptor agonists), alone or combined, on postoperative outcomes. Out of 1180 records initially identified, 15 studies were retained. Evidence for the benefits of unimodal interventions was limited. Preoperative multimodal programs, including nutrition and physical activity with or without psychological support, showed improvement in postoperative physical performance, muscle strength, and quality of life in patients with esophagogastric and colorectal cancers. However, there was no benefit for postoperative complications, hospital length of stay, hospital readmissions, and mortality. No trial evaluated the impact of fecal microbiota transplantation or oral ghrelin receptor agonists. Further studies are needed to confirm our findings, identify patients who are more likely to benefit from surgical prehabilitation, and harmonize interventions.

## 1. Introduction

Gastrointestinal (GI) cancers refer to malignancy of the GI tract. In incidence order, the five major GI cancers are colorectal, gastric, hepatic, esophageal, and pancreatic. They represent over 25% of total cancer incidence with an overall related mortality of 35%, i.e., 5.0 million new cases and 3.5 million deaths worldwide in 2020 [1,2].

There is a direct relationship between GI cancers and diet. The tumor itself and cancer treatments may induce various symptoms affecting nutritional intake, such as loss of appetite, dry mouth, taste and smell alterations, swallowing problems, abdominal pain, nausea, vomiting, diarrhea, constipation, and bowel obstruction [3]. These symptoms significantly impact food intake and, therefore, nutritional status [4]. Consequently, the reported prevalence of malnutrition is high in patients with GI cancers, ranging from 10 to 70% depending on the stage and tumor location [5,6].

Malnutrition is defined by the Global Leadership Initiative on Malnutrition (GLIM) as an association between low body mass index (BMI), reduced muscle mass or unintentional weight loss, and reduced food intake/assimilation or inflammation [7]. The consequences of malnutrition are substantial, with decreased quality of life, oncologic treatment tolerance, and increased physical impairment and mortality [8]. In patients undergoing GI cancer surgery, malnutrition influences surgical outcomes with increased postoperative complications, hospital length of stay (LOS), hospital readmissions, and postoperative mortality [9,10].

Given the growing burden of GI cancers and cancer-associated malnutrition, there is a significant interest in optimizing surgical management. Indeed, surgery is currently the primary curative treatment for GI cancers. About 65% of colorectal cancer patients, 20% of gastric, hepatic, and esophageal cancer patients, and 10% of pancreatic cancer patients will undergo elective surgery at some point in their cancer pathway [11]. In this context, prehabilitation programs have recently been integrated into clinical practice. Prehabilitation could be defined as “a process on the continuum of care that occurs between the time of cancer diagnosis and the beginning of acute treatment” [12]. Prehabilitation prior to surgery aims to optimize preoperative global health status to improve postoperative outcomes such as functional capacity, quality of life (QoL), complications, LOS, and mortality [13,14]. These programs usually include nutritional, physical activity, and/or psychological interventions [15]. Despite promising results, the latest Enhanced Recovery After Surgery (ERAS) guidelines for perioperative care in elective colorectal surgery and esophagectomy has mentioned a low level of evidence of prehabilitation programs [16,17]. Indeed, the efficiency of these interventions has been debated in previous systematic reviews of GI cancers, and standardization in terms of the clinical pathway is needed [18,19,20]. Recent innovations in nutritional therapy could also be discussed as part of such prehabilitation programs. Indeed, there is a growing interest in the modulation of gut microbiota through probiotics, symbiotics, or fecal microbiota transplantation to improve nutritional status, and recent studies suggest a link between malnutrition and fecal microbiota in colorectal cancer patients [21,22,23]. Finally, pharmaceutical molecules such as anamorelin could also contribute to prehabilitation programs. This ghrelin receptor agonist with appetite-enhancing effects has recently been shown to increase body weight in malnourished patients with GI cancers [24,25].

This article aims to (1) review systematically the latest evidence of the effects of different preoperative prehabilitation interventions (nutrition, physical activity, probiotics and symbiotics, fecal microbiota transplantation, and ghrelin receptor agonists, alone or combined) on postoperative outcomes after GI cancer surgery and, (2) discuss new potential therapeutic targets as part of prehabilitation in patients with GI cancers.

## 2. Materials and Methods

This systematic review followed the Preferred Reporting Items for Systematic Reviews and Meta-Analyses (PRISMA) 2020 guidelines [26].

### 2.1. Eligibility Criteria

Randomized controlled trials (RCTs) published in the last five years (between 1 January 2017 and 5 August 2022) were considered for inclusion if they met the following criteria:Population: adults with a malignant GI cancer (esophagus, stomach, duodenum, jejunum, ileum, colon, rectum, anal, liver, pancreas) scheduled for elective surgery.Intervention: preoperative prehabilitation intervention(s), alone or combined, for a minimum of 7 days:
○Nutrition: protein supplementation, oral nutritional supplements (ONS), enteral nutrition (EN), parenteral nutrition (PN);○Physical activity: resistance, endurance, balance, flexibility exercises;○Pro- or symbiotic supplementation;○Fecal microbiota transplantation;○Oral ghrelin receptor agonists.Comparison: comparison group of no intervention, placebo, or other preoperative prehabilitation intervention(s).Outcomes: postoperative muscle mass, muscle strength, physical performance, QoL, surgical and general postoperative complications, LOS, hospital readmission and/or mortality.

Observational studies, study protocols, and reviews not in the English language were excluded. Moreover, studies were excluded if RCTs included participants with no GI malignant disease and if the intervention was perioperative or postoperative only. In addition, nutritional advice alone, nutrient supplementation other than protein, micronutrient supplementation, prebiotics, alternative medicine, and inspiratory muscle training were not considered. Finally, RCTs were not retained if their outcomes did not meet those selected in this review or were assessed only preoperatively.

### 2.2. Information Sources and Search Strategy

The authors used the MEDLINE electronic database (Pubmed) to identify eligible articles. After discussion, the following filter was determined: (“Gastrointestinal” OR “Digestive” OR “Esophageal” OR “Oesophagus” OR “Gastroesophageal” OR “Esophagogastric” OR “Gastric” OR “Stomach” OR “Intestinal” OR “Small intestine” OR “Duodenum” OR “Jejunum” OR “Ileum” OR “Caecum” OR “Colon” OR “Colorectal” OR “Rectosigmoid” OR “Rectum” OR “Rectal” OR “Anal” OR “Gut” OR “Liver” OR “Hepatic” OR “Hepato*” OR “Pancreas” OR “Pancreatic” OR “Intestin” OR “Bowel”) AND (“Cancer*” OR “Tumor*” OR “Adenocarcinoma” OR “Carcinoma” OR “Malignant Neoplasm” OR “Malignancy” OR “Neoplasm*”) AND (“Surgery” OR “Operative” OR “Resection” OR “Surgical”) AND (“Function*” OR “Performance” OR “Muscle*” OR “Strength*” OR “Fat-free mass” OR “Lean body mass” OR “Lean tissue mass” OR “Sarcopenia” OR “Cachexia” OR “Quality of life” OR “Mortality” OR “Death*” OR “Complication*” OR “Length of stay” OR “Readmission*”) AND (“Controlled trial*” OR “Random*”) AND (“Nutrition” OR “Enteral nutrition” OR “Parenteral nutrition” OR “Oral Nutritional Supplement*” OR “Oral Nutritive Supplement*” OR “Dietary supplement*” OR “Protein” OR “Amino acid*” OR “Physical activity” OR “Exercise” OR “Endurance” OR “Resistance” OR “Aerobic” OR “Fecal microbiota transplant*” OR “Fecal transplant*” OR “Stool transplant*” OR “Anamorelin” OR “Ghrelin receptor agonist” OR “Probiotic*” OR “Symbiotic*” OR “Prehabilitation”).

The MEDLINE electronic database was last consulted on 5 August 2022.

### 2.3. Study Selection Process

Two reviewers (J.M. and A.H.) independently screened the titles and abstracts of the eligible articles. If the inclusion criteria were met, the full texts were reviewed to evaluate final inclusion in the systematic review. At each step of the study selection process, any differences were reviewed by J.M. and A.H., and a consensus decision was reached.

### 2.4. Data Collection Process and Data Items

Among the included studies, the data extraction was equally divided between the two reviewers (J.M. and A.H.). The following data were collected from each retrieved article: main population characteristics (sample size, country, tumor characteristics and stage, malnutrition risk, sex, age, BMI, type of surgery, surgical approaches, operative time, and length of stay), intervention (timepoint, duration, description, and sample size), postoperative outcomes (muscle mass, muscle strength, physical performance, QoL, surgical and general complications, LOS, hospital readmission, and/or mortality). The reviewers also recorded the methods used to assess outcomes, main results, and limitations.

### 2.5. Study Risk of Bias Assessment

The risk of bias was assessed with Version 2 of the Cochrane risk-of-bias tool for randomized trials (RoB2) [27]. Five domains of bias (22 items) are considered in this tool: bias arising from the randomization process, bias due to deviations from intended interventions, bias due to missing outcome data, bias in the measurement of the outcome, and bias in the selection of the reported result. Responses provided for each item are: “yes”, “probably yes”, “probably no”, “no”, and “no information”. The risk-of-bias judgment for each domain is “low”, “some concerns”, and “high”.

## 3. Results

### 3.1. Study Selection

The study selection process is presented in the flow diagram (Figure 1). The literature search identified 1180 records, out of which 1104 were excluded after an initial screening of the title and abstract. Seventy-six articles were considered for eligibility, and after full-text assessment, 61 were excluded. The main reasons for exclusion were related to the type of intervention (perioperative intervention, intervention < 7 days), the study design (non-RCT), and the population (participants without GI cancer). Finally, 15 articles met the criteria for inclusion in the final qualitative analysis.

### 3.2. Study Characteristics

Appendix A presents the main population characteristics. The population was heterogeneous, with participants having various tumor characteristics and stages. Colorectal cancers were the most represented (n = 7 studies), followed by esophagogastric cancers (n = 3), pancreatic cancers (n = 2), hepatic cancers (n = 2), and all types of GI cancers (n = 1). Interestingly, nutritional status at baseline was rarely assessed; if assessed, few patients were at risk of malnutrition or malnourished. Most patients did not receive neoadjuvant treatment. Finally, even for the same tumor location, there was a wide disparity in the type of surgery and surgical approaches between studies.

Table 1, Table 2, Table 3, Table 4 and Table 5 detail the studies included in the systematic review. Most RCTs assessed the effects of unimodal preoperative interventions: nutrition (n = 5, Table 1), physical activity (n = 3, Table 2), and probiotics (n = 2, Table 3). Other studies combined preoperative nutritional and physical activity interventions (n = 2, Table 4) and multimodal interventions (>2 interventions, n = 3, Table 5). No RCT assessed the impact of fecal microbiota transplantation or oral hrelin receptor agonists as preoperative interventions. The duration of interventions varied from 1 to 15 weeks, with an intervention lasting ≤4 weeks in most of the studies. The modalities of interventions differed significantly among studies. Finally, postoperative complications were the most frequently studied outcome, followed by LOS. Muscle mass, strength, physical performance, and QoL have been scarcely studied.

### 3.3. Risk of Bias in the Studies

The Cochrane RoB2 was used to assess the risk of bias in RCTs included in the systematic review [27]. The five domains of bias and overall bias are summarized for each RCT in Table 6. Among the included studies, a majority presented some concerns regarding the overall risk of bias (10 studies out of 15). Four studies were judged at high risk of overall bias, mainly due to deviations from the intended intervention, selection of the reported results, and/or measurement of the outcome. One study was considered at low risk of overall bias.

### 3.4. Main Findings

#### 3.4.1. Unimodal Nutritional Interventions

Five recent RCTs assessed the impact of preoperative nutritional interventions on postoperative outcomes in patients with GI cancers (Table 1).

He et al. compared a 1-week EN intervention to dietary advice alone in patients with malignant gastric cancer [28]. Most patients were malnourished. They did not find a significant difference in postoperative complications and 1-month hospital readmissions between groups. One study evaluated the effect of a 1-week high protein ONS supplementation on postoperative muscle mass and strength, complications, and LOS in patients with colorectal cancers [29]. They failed to demonstrate any difference between groups for all postoperative outcomes. Similar results were found in two other studies. Preoperative immunonutrient-enriched and eicosapentaenoic acid (EPA) enriched ONS did not improve postoperative complications, LOS, and hospital readmission rates in patients with colon and periampullary cancers [30,32]. Finally, in hepatic cancer patients undergoing surgery, Okabayashi et al. reported no difference in postoperative complications and 90-day mortality after 2-weeks of oral L-carnitine supplementation [31]. However, postoperative liver function was more rapidly restored in the intervention group. Interestingly, they reported a shorter median LOS in the intervention group compared to the control group (10 [7–157] days vs. 12 [5–144] days, *p* = 0.048).

To summarize, a 1-week nutritional prehabilitation had no benefits on muscle mass, muscle strength, postoperative complications, LOS, and hospital readmission rates. Amino-acid supplementation before surgery, during at least 2 weeks, may reduce LOS after hepatectomy in patients with liver cancer.

#### 3.4.2. Unimodal Physical Activity Interventions

Three studies assessed the effects of physical activity interventions before surgery in patients with GI cancers (Table 2).

Berkel et al. evaluated the impact of a 3-week preoperative supervised aerobic and resistance exercise program in colorectal cancer or premalignant colorectal lesions [33]. Patients at high risk for postoperative complications with low preoperative physical fitness (i.e., oxygen consumption, VO_2_) at the ventilatory anaerobic threshold (VAT) < 11 mL/kg/min) were included. Interestingly, prehabilitation significantly reduced 30-day overall postoperative complications compared to the control group (relative risk: 0.59 [95% CI 0.37 to 0.96]; *p* = 0.024). This benefit was associated with a 10% improvement in preoperative physical performance after the 3-week program in the intervention group (VO_2_ at VAT: +0.97 mL/kg/min, [95% CI 0.3 to 1.6]; *p* = 0.006). However, there was no significant difference between the two groups regarding postoperative complications, LOS, and readmissions within 30 and 90 days after surgery.

In two smaller studies, the results were inconclusive regarding the benefits of a 2 to 6 weeks preoperative aerobic, resistance, and respiratory exercise intervention in patients with different types and stages of GI cancers [34,35]. The exercise programs were considered safe and tolerable, but postoperative complications and LOS were not significantly different between the intervention and the control groups. In addition, prehabilitation did not improve muscle strength, physical performance, and QoL in the preoperative period.

Although exercise before elective surgery could reduce postoperative complication rates, evidence for a positive impact of preoperative physical activity intervention on postoperative outcomes is limited.

#### 3.4.3. Unimodal Probiotics and Symbiotics Interventions

Probiotics or symbiotics interventions as part of prehabilitation in colorectal and hepatic cancer patients showed discordant findings (Table 3).

In patients with hepatic cancer, Roussel et al. found no significant difference in all postoperative complications and mortality after two weeks of oral probiotics versus placebo [36].

Polakowski et al. compared a 1-week symbiotic intervention versus a placebo in patients with colorectal cancer [37]. They found no difference in non-infectious postoperative complications and 30-day mortality. However, they reported a lower rate of postoperative infectious complications in the intervention group compared to the control group (2.8% vs. 18.9%; *p* = 0.02) and a reduced median LOS (3 [3–5] days vs. 4 [3–21] days; *p* < 0.001).

Thus, few RCTs investigated the impact of preoperative probiotics/symbiotics in GI cancer patients, and the benefits remain controversial.

#### 3.4.4. Combined Nutritional and Physical Activity Interventions

Two RCTs assessed a bimodal intervention combining nutrition and physical activity in patients with GI cancers undergoing surgery (Table 4).

In patients with pancreatic cancer and cholangiocarcinoma, Ausania et al. compared a prehabilitation program, including high-intensity aerobic training and nutritional supplementation, for at least one week prior to surgery versus the standard of care [38]. They reported no significant difference between groups in postoperative complications, LOS, and hospital readmissions. Muscle strength and physical performance were only available for the intervention group before surgery. Nevertheless, the authors observed a slight preoperative improvement of respectively 16% and 21% in the right and left handgrip strength and 19% in 10 m walk test.

In non-metastatic esophagogastric cancer patients, home-based moderate aerobic and resistance exercises combined with whey protein supplementation showed a significant improvement in physical performance compared to usual care [39]. After the prehabilitation period, preoperative and postoperative 6-minute walking distances were better in the intervention group compared to the control group (respectively, mean changes from baseline: 36.9 ± 51.4 m vs. −22.8 ± 52.5 m; *p* < 0.001 and 15.4 ± 65.6 m vs. −81.8 ± 87.0 m; *p* < 0.001). Moreover, 62% of patients experienced an improvement in preoperative physical performance after prehabilitation versus 4% in the control group (*p* < 0.001) and 52% postoperatively versus 6%, respectively (*p* < 0.001). No differences were reported between groups regarding postoperative complications, LOS, hospital readmissions, and mortality.

Prehabilitation programs combining nutrition and physical activity did not appear to affect postoperative complications, LOS, hospital readmissions, and mortality in pancreatic and esophagogastric cancer patients. However, prehabilitation may improve pre- and post-operative physical performance in patients with esophagogastric malignancies.

#### 3.4.5. Multimodal Interventions (>2 Interventions)

The effects of preoperative multimodal programs, including nutritional, exercise, and psychological interventions, have been tested in patients with esophagogastric and colorectal cancers (Table 5).

In patients with locally advanced esophagogastric cancer, a 15-week multimodal prehabilitation program helped maintain postoperative muscle strength and QoL [40]. Before neoadjuvant chemoradiotherapy and until surgery, the intervention consisted of aerobic, resistance, and flexibility exercises, nutritional support, and psychological coaching. Six weeks after surgery, the handgrip strength was higher in the prehabilitation group than in the usual care group (mean changes from baseline: 96% [95% CI 88 to 104] vs. 87% [95% CI 75 to 99]; *p* = 0.009). In addition, the global QoL score was better in prehabilitation subjects than controls after two weeks (*p* = 0.001), six weeks (*p* = 0.001) and six months (*p* = 0.003) post-surgery. Interestingly, the authors also demonstrated that prehabilitation was associated with better preservation of skeletal muscle index after neoadjuvant treatments (mean changes from baseline: −11.6 cm^2^/m^2^ [95% CI −14.2 to −9.0] vs. −15.6 cm^2^/m^2^ [95% CI −18.7 to −15.4]; *p* = 0.049), and VO_2_ peak (mean change: −0.4 mL/kg/min [95% CI −0.8 to 0.1] vs. −2.5 mL/kg/min [95% CI −2.8 to −2.2]; *p* = 0.022). However, these outcomes were not assessed postoperatively.

Carli et al. were interested in the impact of a 4-week aerobic, resistance, and flexibility training in addition to whey protein supplementation and personalized coping strategies [41]. Frail patients (Fried frailty index > 1) were randomized to this intervention either before or after surgery. Postoperative complications, LOS, and readmissions within 30 days after surgery were not significantly different between the prehabilitation and the postoperative rehabilitation groups.

Finally, Minnella et al. compared two different exercise training protocols as part of a multimodal prehabilitation program [42]. Patients with non-metastatic colorectal cancer were randomized to supervised high or moderate-intensity interval training combined with whey protein supplementation and relaxation training for four weeks. Both protocols enhanced preoperative physical performance, assessed by VO_2_ at VAT, with no difference between groups (high-intensity interval training 1.97 mL/kg/min [95% CI 0.75 to 3.19] vs. moderate intensity interval training 1.71 mL/kg/min [95% CI 0.56 to 2.85]; *p* = 0.753). Two months after surgery, the high-intensity interval training group had better physical fitness than the moderate-intensity interval training group (mean changes of VO_2_ at VAT from baseline: 2.36 mL/kg/min [95% CI 0.38 to –4.34]; *p* = 0.021).

Multimodal prehabilitation in patients with locally advanced esophagogastric or colorectal cancer could improve postoperative muscle strength, physical performance, and QoL. Nevertheless, recent RCTs failed to demonstrate its benefits on postoperative complications, LOS, hospital readmissions, and mortality.

## 4. Discussion

This systematic review synthesizes the results of RCTs on the impact of surgical prehabilitation on different postoperative outcomes in patients with GI cancers over the past five years. Study populations and intervention modalities were heterogeneous, and most RCTs were with some concerns or at high risk of bias. Thus, the results of our systematic review should be considered with caution. Evidence for the benefits of unimodal nutritional, physical activity, or probiotics/symbiotics interventions is limited and discordant. Multimodal programs, which combined nutrition and physical activity with or without psychological support, showed improvement in postoperative physical performance, muscle strength, and QoL in patients with esophagogastric and colorectal cancers. However, there was no benefit for postoperative complications, LOS, hospital readmissions, and mortality. Finally, no RCT evaluated the impact of fecal microbiota transplantation or oral ghrelin receptor agonists as preoperative interventions. This raises several questions regarding population, type of prehabilitation programs, and outcome assessments.

### 4.1. Variability of Population and Type of Surgery

Firstly, nutritional status was not systematically assessed before interventions, and screening tools were inconsistent. When the information was provided, only a few patients were at risk of malnutrition or malnourished. However, the European Society for Clinical Nutrition and Metabolism (ESPEN) recommends screening of malnutrition risk for all patients within 48 h after hospital admission. In addition, preoperative nutritional intervention is indicated for patients at high risk of malnutrition or malnourished [43,44]. A recent pooled analysis confirmed that nutritional status influences the efficiency of a 4-week multimodal prehabilitation in colorectal cancer surgery [45]. Thus, prehabilitation is very likely to benefit mostly malnourished patients, which are at higher risk of postoperative complications [46,47,48]. Secondly, the reviewed studies focused on a wide variety of surgical interventions. The type of tissue or organ resected has an impact on potential surgical outcomes and risk of complications [49]. For instance, esophageal or pancreatic surgeries have a higher rate and severity of complications than a right hemicolectomy or limited gastric resection [50]. Furthermore, the type of surgical approach impacts postoperative outcomes: minimally invasive approaches, either laparoscopic or robotic, have shown significant improvement in postoperative outcomes through a reduction of surgical trauma in GI cancers [51]. However, the penetration of minimally invasive techniques is variable depending on tumor types and expertise. For example, minimally invasive colon surgery is now the gold standard in most centers in developed countries, while minimally invasive oncologic hepato-pancreato-biliary surgery is only performed in high-volume centers of expertise [52]. In this review, the benefit of surgical prehabilitation concerned upper GI, colorectal, and hepatic cancers, large sample sizes, and intervention of at least 2 weeks. As observed in daily clinical activity, the impact of prehabilitation seems more obvious in patients with good prognoses, often treated through a minimally invasive procedure, as opposed to patients with poor prognosis cancers undergoing major laparotomies and more debilitating resections. Targeting these populations with nutritional issues could thus lead to more clinically meaningful and cost-effective interventions.

### 4.2. Variability of Surgical Prehabilitation Programs

Currently, there is no clear consensus regarding the modalities of prehabilitation programs before GI cancer surgery. The high heterogeneity of interventions in this review illustrates this issue. Regarding nutrition, ERAS, and ESPEN guidelines suggests nutritional support for 7–14 days in patients not meeting preoperative energy needs [44,53]. Despite a strong consensus among experts, there is currently a lack of scientific evidence. In this review, most negative studies had a nutritional intervention of only one week [28,29,30,32,38]. This suggests that such intervention may be too short to impact postoperative outcomes of GI cancers. The role of immunonutrition as part of prehabilitation could also be discussed. Indeed, immunonutrition is recommended for 5–7 days preoperatively, mostly in upper GI cancer patients [54]. In this review, we did not find benefit of a 1-week preoperative immunonutrition supplementation in colon cancer [30]. However, few patients were at risk of malnutrition, and adherence to the intervention was not mentioned. For physical activity interventions, professionals agree on the importance of aerobic and resistance exercises, but there are no well-defined patterns. This is reflected in this article with as many different exercise protocols as studies. The World Health Organization (WHO) recommends 150–300 min of aerobic exercise at moderate intensity each week and two sessions of resistance exercise for healthy adults [55]. These guidelines should be adapted to the physical abilities of patients with GI cancer to ensure safety and efficiency. Furthermore, the benefits of psychological interventions as part of the preoperative pathway are well recognized. However, there is no consensus regarding the type, timing, and frequency of preoperative interventions used in oncological patients [56]. Moreover, psychological support is not systematically integrated into prehabilitation programs. Only three studies considered psychological care in our review [30]. Finally, some authors highlighted the importance of multimodal instead of unimodal prehabilitation to maximize the impact on postoperative outcomes [57]. Our results support this statement in patients with esophagogastric and colorectal cancers. To conclude, the harmonization of multimodal prehabilitation programs is needed with guidelines personalized and adapted to the specificity of surgical patients with GI cancers.

### 4.3. Challenges of Surgical Prehabilitation

Prehabilitation interventions are limited by the patient’s context. A proven correlation exists between poor lifestyle habits and the most common GI cancers [58]. Added to the psychological shock related to the cancer diagnosis, these habits do not necessarily create the ideal conditions for adherence to major lifestyle modifications such as nutritional and psychological support and/or increased physical activity. A major challenge in GI cancer prehabilitation is also the surgical timing, as optimizing a patient’s global condition requires sufficient time. Most study interventions in this review ranged from 1 to 3 weeks. With such limited exposure, these interventions are less likely to demonstrate benefits on early postoperative outcomes. Our findings support that at least two weeks of surgical prehabilitation are necessary to expect an impact [28,29,30,32,38]. Unfortunately, in GI cancer resections, time is often considered of the essence to avoid disease progression, although this concept is challenged [59,60]. Furthermore, the main drivers of good postoperative surgical outcomes are usually the healing of anastomoses and wounds, bowel motility, and return to normal physical activities in GI oncologic resections. Markers of malnutrition and low physical fitness have been correlated with the impairment of postoperative surgical outcomes [61]. Thus, the assessment of postoperative nutritional markers is essential to identify the benefits of prehabilitation and induce changes in clinical practice. Finally, multimodal interventions can have advantages beyond the scope of strict surgical outcomes, improving QoL and long-term survival, facilitating rehabilitation, and return to activity. As these interventions focus on lifestyle modifications, they could have long-standing effects not necessarily detected within the timeframe of these studies. A recent pool analysis showed an improvement in 5-year disease-free survival after prehabilitation for colorectal cancer surgery [62]. Other long-term outcomes such as incisional hernia, abdominal wall, and digestive or sexual functions are also rarely investigated. These outcomes could potentially be improved by surgical prehabilitation directed towards reinforcing the quality of tissues and physiological reserve at the time of surgery [63].

### 4.4. Prospects for New Therapeutic Strategies

Inconclusive results of RCTs evaluating surgical prehabilitation programs in GI cancers could also be explained by the multifactorial origin of malnutrition in cancer patients [64]. Therefore, new potential therapeutic targets should be considered. This review identified two RCTs with probiotics/symbiotics and no RCTs with fecal microbiota transplantation nor oral ghrelin receptor agonists. However, there is a growing interest in using probiotics or fecal microbiota transplantation to modulate the gut microbiota because its composition seems to play a role in cancer development and response to treatment [65]. For example, *Fusobacterium nucleatum* is abundant in colorectal cancer patients with recurrence after chemotherapy and is associated with resistance to treatment [66]. Other microbiota species, namely *Atopobium vaginae*, *Selenomonas sputigena*, and *Faecalibacterium prausnitzii*, have been identified as diagnostic biomarkers of malnutrition and poor prognosis in colorectal cancer [21], whereas *Clostridium butyricum* may have great potential for improving the nutritional status of malnourished patients [22]. Concerning probiotics, a 2-week probiotic intervention in patients with hepatocellular carcinoma did not improve postoperative complications and mortality [36]. Nevertheless, infectious complications and LOS were significantly lower in patients with colorectal cancer receiving a 1-week symbiotic intervention [37]. Regarding fecal microbiota transplantation and ghrelin receptor agonists, no studies evaluate these innovative strategies in GI cancer surgical prehabilitation. To date, one phase II RCT evaluated the potential of fecal microbiota transplantation from overweight or obese donors to malnourished patients with advanced gastroesophageal cancer prior to palliative chemotherapy. Fecal microbiota transplantation did not affect malnutrition but could improve response and survival [23]. Finally, a selective agonist of the ghrelin/growth hormone secretagogue receptor, anamorelin (ONO-7643), may be interesting in fighting preoperative malnutrition. This molecule exhibits appetite-stimulating and anabolism properties. Phase II and III studies showed that a daily oral dose of 100 mg, compared to a placebo, increased appetite, total body weight, and lean body mass and improved the functional, physical, mental, and psychological state of malnourished patients with non-small-cell lung cancer [67,68,69]. Despite these encouraging results, the European Medicines Agency has not authorized the marketing of anamorelin for malnutrition treatment. Indeed, those studies did not affect the patient’s QoL, muscle strength, and physical performance. In GI cancers, similar results were obtained with an increased appetite and a marginal body weight gain [24,25]. Currently, there is only one multicenter RCT registered in ClinicalTrials.gov (NCT04844970) that evaluates the efficacy and safety of anamorelin in 100 patients with advanced pancreatic cancer. To summarize, there is a lack of evidence for integrating new nutritional therapies as part of surgical multimodal prehabilitation programs. However, the first data available in other settings are quite promising, and these recent innovations could be evaluated in well-designed studies [70].

### 4.5. Strengths and Limitations

Our systematic review presents several strengths. Two reviewers independently screened titles and abstracts of the eligible articles, and only RCTs were included. It provides the latest evidence on surgical prehabilitation in patients with GI cancers and integrates potential new therapeutic strategies. However, there are some limitations. Firstly, the population was heterogeneous across the different RCTs. All GI cancers, despite nutritional status, tumor stage and location, different resections, and surgical approaches, with or without preoperative neoadjuvant therapy, were considered. Secondly, a large variety of interventions and outcome assessment methods were included. Thirdly, there is a lack of data on patient adherence which may affect the effects of the interventions. Furthermore, RCTs generally compared an intervention group to a control group, receiving nutritional and/or physical activity advice (usual standard of care). Consequently, the results might have been affected as the control group could be considered “partially prehabilitated”. Finally, information about the postoperative nutritional and physical management of patients included in the different studies was sparsely reported.

In the absence of consensus for prehabilitation in GI cancer surgery, we suggest considering the following parameters when designing and conducting future studies (Figure 2).

## 5. Conclusions

Current evidence of the impact of unimodal prehabilitation on postoperative outcomes in GI cancer surgery remains unclear. However, postoperative physical performance, muscle strength, and QoL could be improved with surgical multimodal prehabilitation in patients with esophagogastric and colorectal cancers. Further studies are needed to confirm our findings, identify surgical cancer patients more likely to benefit from prehabilitation, harmonize interventions, and integrate new therapeutic strategies.

## Figures and Tables

**Figure 1 cancers-15-01881-f001:**
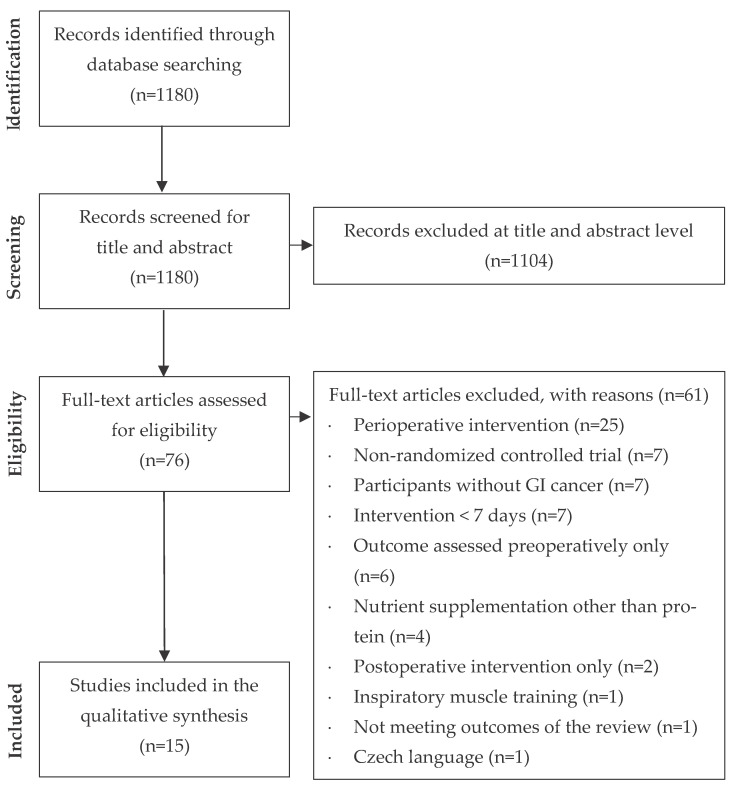
Flow diagram of the study selection process.

**Figure 2 cancers-15-01881-f002:**
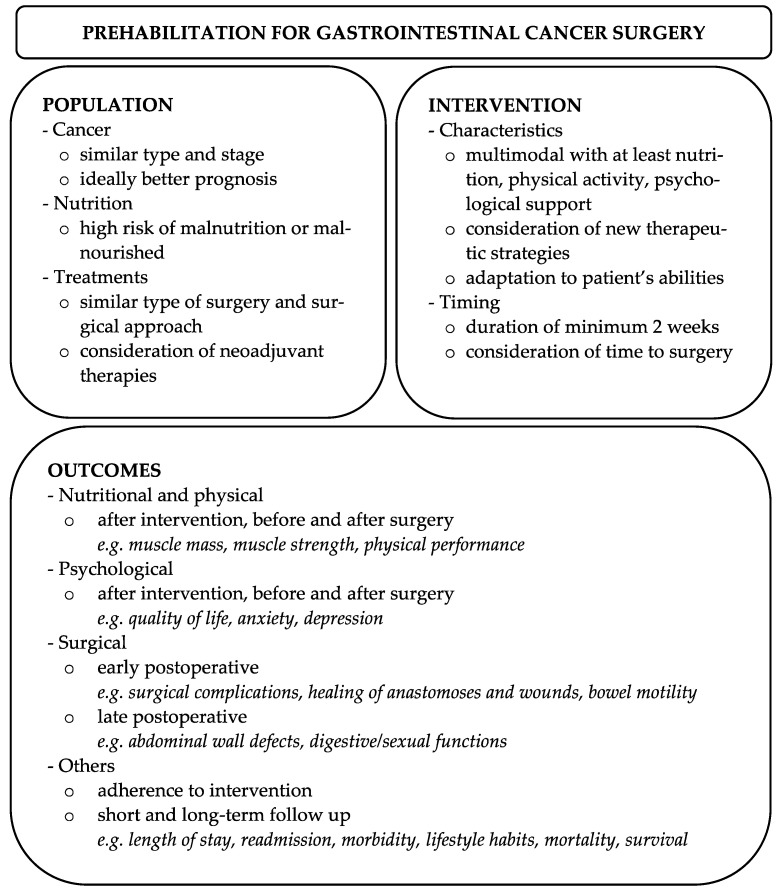
Recommendations for future studies evaluating prehabilitation in GI cancer surgery.

**Table 1 cancers-15-01881-t001:** Nutritional interventions before surgery in patients with gastrointestinal cancers.

Studies	Population	Intervention	Postoperative Outcomes	Results	Limitations
He et al., 2022 [28]	Gastric cancer scheduled for radical gastrectomy n = 67	1 week - INT: EN 500 mL, 450 kcal, 17 g protein/day (n = 32) - CO: dietary advice (n = 35)	- Postoperative complications ^a^ - 30-day hospital readmission	No significant differences in all outcomes between groups	Lack of information on baseline nutritional parameters, important variations of surgical risk levels, no information on surgical approach, short duration of intervention, no intention to treat analysis
Tesar et al., 2022 [29]	Colorectal cancer scheduled for colorectal resection n = 120	1 week - INT: ONS 250 mL, 600 kcal, 24 g protein/day or ONS for diabetics 400 mL, 600 kcal, 30 g protein/day (n = 60) - CO: no ONS (n = 60)	- Muscle mass: BIA - Muscle strength: handgrip strength - 30-day postoperative complications - LOS	No significant differences in all outcomes between groups	Few patients at risk of malnutrition, lack of information on baseline nutritional parameters, important variations of surgical risk levels, short duration of intervention, adherence to the intervention not mentioned, no intention to treat analysis
Lee et al., 2021 [30]	Colon cancer scheduled for colon resection n = 176	1 week - INT: ONS 400 mL, 400 kcal, 20 g protein, 1 g arginine, 0.92 g ω-3 fatty acids/day (EPA + DHA + ALA) (n = 88) - CO: normal diet (n = 88)	- 30-day postoperative complications - LOS - 30-day hospital readmission	No significant differences in all outcomes between groups	Few patients at risk of malnutrition, lack of information on baseline nutritional parameters, important variations of surgical risk levels, short duration of intervention, adherence to the intervention not mentioned, no intention to treat analysis
Okabayashi et al., 2020 [31]	Hepatic cancer scheduled for hepatectomy without hepaticojejunostomy n = 208	2 weeks - INT: Oral L-carnitine, 30 mg/kg/day (n = 102) - CO: usual intake (n = 106)	- Postoperative complications ^a^ - LOS - 90-day mortality	- No significant differences in postoperative complications and mortality between groups - LOS significantly shorter in the L-carnitine group compared to the control group	No information on tumor stage and baseline nutritional parameters, important variations of surgical risk levels, no information on surgical approach, short duration of intervention, adherence to the intervention not mentioned
Ashida et al., 2019 [32]	Periampullary cancer scheduled for pancreatoduodenectomy n = 24	1 week - INT: ONS 500 mL, 600 kcal, 32 g protein, 2 g EPA/day (n = 12) - CO: ONS 500 mL, 600 kcal, 30 g protein/day (n = 12)	- Postoperative complications ^a^	No significant difference in postoperative complications between groups	Small sample size, no information on tumor stage, lack of information on baseline nutritional parameters, short duration of intervention, adherence to the intervention not mentioned, no intention to treat analysis

^a^ Timing not mentioned. Abbreviations: INT: Intervention group, CO: Control group, yrs: years, EN: Enteral Nutrition, ONS: Oral Nutritional Supplement, EPA: Eicosapentaenoic acid, DHA: docosahexaenoic acid, ALA: α-linolenic acid, LOS: Length of stay.

**Table 2 cancers-15-01881-t002:** Physical activity interventions before surgery in patients with gastrointestinal cancers.

Studies	Population	Intervention	Postoperative Outcomes	Results	Limitations
Berkel et al., 2022 [33]	Colorectal cancer or premalignant colorectal lesions, high risk for postoperative complications, scheduled for colorectal resection n = 74	3 weeks - INT: supervised aerobic and resistance exercise, 60 min, 3x/week (n = 39) - CO: nutritional counseling and advice on smoking cessation (n = 35)	- 30-day postoperative complications - LOS - 30 and 90-day hospital readmission	- Overall postoperative complications significantly lower in the intervention group compared to the control group - No differences in the type of complications, LOS, and hospital readmissions between groups	Lack of information on baseline nutritional parameters, important variations of surgical risk levels, no information on tumor stage, many patients excluded after randomization, adherence to the intervention not mentioned
Steffens et al., 2021 [34]	Gastrointestinal cancer scheduled for pelvic exenteration or cytoreductive surgery & hyperthermic intraperitoneal chemotherapy n = 22	2 to 6 weeks - INT: supervised aerobic, resistance, and respiratory exercise, 60 min, 1x/week + home-based aerobic, resistance, and respiratory exercise, 60 min, 4x/week + walking, ≥30 min, 7x/week (n = 11) - CO: nutritional counseling, advice on smoking cessation, and reduction of alcohol intake (n = 11)	- Postoperative complications ^a^ - LOS	No significant differences in all outcomes between groups	Low recruitment rate, small sample size, no information on tumor stage and baseline nutritional parameters, important variations of surgical risk levels, high loss to follow-up rate in intervention group, feasibility analysis, no intention to treat analysis
Karlsson et al., 2019 [35]	Colorectal cancer scheduled for colorectal resection n = 23	2 to 6 weeks - INT: home-based supervised aerobic, resistance, and respiratory exercise, 60 min, 2–3/week (n = 11) - CO: usual care, advice for 150 min/week of moderate physical activity (n = 12)	- 30-day postoperative complications - LOS	No significant differences in all outcomes between groups	Small sample size, difference in baseline characteristics between groups, no information on baseline nutritional parameters, important variations of surgical risk levels, feasibility analysis, no intention to treat analysis

^a^ Timing not mentioned. Abbreviations: INT: Intervention group, CO: Control group, LOS: Length of stay, QoL: Quality of life.

**Table 3 cancers-15-01881-t003:** Probiotics and symbiotics interventions before surgery in patients with gastrointestinal cancers.

Studies	Population	Intervention	Postoperative Outcomes	Results	Limitations
Roussel et al., 2022 [36]	Hepatocellular carcinoma with underlying cirrhosis scheduled for liver resection n = 54	2 weeks - INT: oral probiotics—10^9^ concentration of 5 lactic acid bacteria, 2x/day (n = 27) - CO: corn starch placebo (n = 27)	- 90-day postoperative complications - 90-day mortality	No significant differences in all outcomes between groups	Differences in baseline characteristics between groups, no information on tumor stage and baseline nutritional parameters, important variations of surgical risk levels, no information on surgical approach
Polakowski et al., 2019 [37]	Colorectal cancer scheduled for colorectal resection n = 120	1 week - INT: oral symbiotic—6 g of fructooligosaccharide + 10^9^ concentration of 5 lactic acid bacteria, 2x/day (n = 36) - CO: corn starch placebo (n = 37)	- 30-day infectious and non-infectious postoperative complications - LOS - 30-day mortality	- Infectious complications and median LOS significantly lower in the intervention group compared to the control group - No significant difference in non-infectious complications and mortality between groups	Lack of information on baseline nutritional parameters, important variations of surgical risk levels, no information on surgical approach, short duration of intervention, adherence to the intervention not mentioned

Abbreviations: INT: Intervention group, CO: Control group, LOS: Length of stay.

**Table 4 cancers-15-01881-t004:** Combined nutritional and physical activity interventions before surgery in patients with gastrointestinal cancers.

Studies	Population	Intervention	Postoperative Outcomes	Results	Limitations
Ausania et al., 2019 [38]	Pancreatic or peripancreatic malignancy scheduled for Whipple procedure n = 40	At least 1 week - INT: supervised high-intensity aerobic exercise, 60 min, 5x/week + daily home-based functional exercises + nutritional support (oral and vitamin supplements, total parenteral nutrition if required) (n = 18) - CO: usual care (n = 22)	- Postoperative complications ^a^—LOS - Readmission rate	No significant differences in all outcomes between groups	Small sample size, no information on tumor stage, lack of information on baseline nutritional parameters, short duration of intervention, no information on nutritional support composition, adherence to the intervention not mentioned, no intention to treat analysis
Minnella et al., 2018 [39]	Esophagogastric cancer scheduled for esophagectomy or total or partial gastric resection n = 68	Median length 36 days (IQR 17–73) - INT: Home-based aerobic exercise, 30 min, 3x/week + resistance exercise, 30 min, 1x/week + whey protein (aim protein intake 1.2–1.5 g/kg/d) (n = 34) - CO: usual care (n = 34)	- Physical performance: 6MWD - 30-day postoperative complications - LOS - 30-day hospital readmission - Mortality	- Better physical performance in the intervention group compared to the control group - No significant difference in postoperative complications, LOS hospital readmission, and mortality between groups	Few patients at risk of malnutrition, lack of information on baseline nutritional parameters, important variations of surgical risk levels, no minimal and consistent duration of intervention, no intention to treat analysis

^a^ Timing not mentioned. Abbreviations: INT: Intervention group, CO: Control group, LOS: Length of stay, 6MWD: 6-minute walking distance.

**Table 5 cancers-15-01881-t005:** Multimodal interventions before surgery in patients with gastrointestinal cancers.

Studies	Population	Intervention	Postoperative Outcomes	Results	Limitations
Allen et al., 2022 [40]	Locally advanced esophagogastric cancer undergoing neoadjuvant chemoradiotherapy scheduled for esophagectomy or gastrectomy n = 54	15 weeks—Started before neoadjuvant chemoradiotherapy and continued until surgery: - INT: supervised aerobic, resistance, and flexibility exercise, 60 min, 2x/week + home-based resistance and flexibility exercise, 60 min, 3x/week + nutritional support to cover protein and energy needs + 6 sessions of psychological coaching (n = 26) - CO: usual care (n = 28)	- Muscle strength: handgrip strength - QoL: EORTC QLQ-C30 - Postoperative complications ^a^ - LOS - 30-day hospital readmission - 3-year mortality	- Better postoperative handgrip strength and QoL in the intervention group compared to the control group - No significant difference in postoperative complications, LOS, hospital readmission, and mortality	Lack of information on baseline nutritional parameters, important variations of surgical risk levels, no information on surgical approach, high lost to follow-up rate, data not available for all outcomes in all patients
Carli et al., 2020 [41]	Frail colorectal cancer scheduled for colorectal resection n = 120	4 weeks - INT1 started before surgery: supervised aerobic, resistance, and flexibility exercise, 60 min, 1x/week + home-based aerobic exercise, 30 min, 7x/week and resistance exercise, 3x/week + whey protein (aim protein intake 1.5 g/kg/d) + personalized coping strategies 3x/week (n = 60) - INT2, started after hospital discharge: Same intervention as INT1 (n = 60)	- 30-day postoperative complications - LOS - 30-day hospital readmission	No significant differences in all outcomes between groups	Differences in baseline characteristics between groups, lack of information on baseline nutritional parameters, important variations of surgical risk levels, absence of control group with usual care, poor adherence in the int2 group
Minnella et al., 2020 [42]	Colorectal cancer scheduled for colorectal resection n = 42	4 weeks - INT1: supervised high-intensity interval training (aerobic exercise) and resistance exercise, 40 min, 3x/week + whey protein (aim protein intake 1.5 g/kg/d) + training to relaxation technique (n = 21) - INT 2: same intervention as INT1 but at moderate intensity	- Physical performance: aerobic fitness (V02 at the ventilatory anaerobic threshold) within one and two months after surgery - 30-day postoperative complications - LOS	- At 2 months after surgery, significant improvement in physical performance in the high-intensity interval training group compared to the moderate-intensity interval training group - No difference in postoperative complications and LOS between the two groups	Few patients at risk of malnutrition, lack of information on baseline nutritional parameters, important variations of surgical risk levels, absence of control group with usual care, high loss to follow-up rate, data for postoperative outcome available for only 50% of participants

^a^ Timing not mentioned. Abbreviations: INT: Intervention group, CO: Control group, QoL: Quality of life, LOS: Length of stay, INT1: Intervention N°1, INT2: Intervention N°2, EORTC QLQ-C30: European Organization for the Research and Treatment of Cancer Quality of Life Questionnaire—30 item.

**Table 6 cancers-15-01881-t006:** Version 2 of the Cochrane risk-of-bias tool (RoB2) for randomized controlled trials included in the systematic review.

	Randomization Process	Deviations from the Intended Intervention	Missing Outcome Data	Measurement of the Outcome	Selection of the Reported Result	Overall
**Unimodal nutritional interventions**						
He, 2022 [28]	+	+	+	+	!	!
Tesar, 2022 [29]	+	-	+	-	!	-
Lee, 2021 [30]	+	!	-	!	!	-
Okabayashi, 2020 [31]	+	!	+	!	!	!
Ashida, 2019 [32]	+	+	+	+	!	+
**Unimodal physical activity interventions**						
Berkel, 2022 [33]	+	+	!	+	!	!
Steffens, 2021 [34]	+	-	-	+	!	-
Karlsson, 2019 [35]	+	+	+	+	!	!
**Unimodal probiotics and symbiotics interventions**						
Roussel, 2022 [36]	+	+	+	+	!	!
Polakowski, 2019 [37]	+	+	+	+	!	!
**Combined nutritional and physical activity interventions**						
Ausania, 2019 [38]	+	-	+	+	!	-
Minnella, 2019 [39]	+	+	!	!	!	!
**Multimodal interventions (>2 interventions)**						
Allen, 2022 [40]	+	+	!	+	!	!
Carli, 2020 [41]	!	+	+	+	+	!
Minnella, 2020 [42]	+	+	!	+	!	!
Legend:						
+	Low risk	!	Some concerns	-	High risk	

## Data Availability

No new data created.

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
