# Peer review of "Surgical Prehabilitation in Patients with Gastrointestinal Cancers: Impact of Unimodal and Multimodal Programs on Postoperative Outcomes and Prospects for New Therapeutic Strategies—A Systematic Review"

_cancers, 2023, doi:10.3390/cancers15061881_

Round 1

Reviewer 1 Report

This systematic review evaluates the impact of preoperative prehabilitation interventions on postoperative outcomes in gastrointestinal cancer surgery patients.

The review found limited evidence for the benefits of single interventions, and multimodal programs including nutrition and physical activity, with or without psychological support.

No impact was found on postoperative complications, hospital length of stay, hospital readmissions, and mortality.

Dr Mareschal and col. did a great work with this extensive work and should be congratulated.

- Author should have discuss the current data regarding prehabilitation's role on survival (OS and RFS or PFS).

Author Response

Thank you very much for your review and your comments. We did not choose to include survival in our outcomes. However, your suggestion is relevant and we added:

  • A reference about survival in the discussion (line 435): « Finally, multimodal interventions can have advantages beyond the scope of strict surgical outcomes, improving QoL and long-term survival, facilitating rehabilitation, and return to activity. As these interventions focus on lifestyle modifications, they could have long-standing effects not necessarily detected within the timeframe of these studies. A recent pool analysis showed an improvement of 5-year disease-free survival after prehabilitation for colorectal cancer surgery [62]. »
  • “survival” under short and long-term follow-up outcomes for recommendations for future studies evaluating prehabilitation in gastrointestinal cancer surgery (Figure 2).

Reviewer 2 Report

This is an interesting study but has significant limitations. I think the assessment of improvement in all the studies uses was different as was the type of GI malignancy. Therefore a more focused approach would be prudent and potentially determining some specific objective marker to show improvement. 

Author Response

Thank you very much for your review and your comments.

Indeed, the assessment methods of half of our outcomes (postoperative muscle mass, muscle strength, physical performance, QoL) tend to vary across studies. We mentioned assessment methods in all the tables to be transparent about this heterogeneity. However, we decided to consider various assessment methods as suggested in different guidelines for malnutrition and sarcopenia (e.g. T. Cederholm. GLIM criteria for the diagnosis of malnutrition: A consensus report from the global clinical nutrition community. Clinical Nutrition. 2018. and A. Cruz-Jentoft. Sarcopenia: revised European consensus on definition and diagnosis. Age Ageing. 2019). We added this limitation in the systematic review (Line 491): “Secondly, a large variety of interventions and outcome assessment methods were included”.

Regarding the type of gastrointestinal cancers, the inclusion of all types could be seen as a limitation, but we chose to give an overview of the effect of prehabilitation in this population. We discussed this limitation in the discussion (Lines 368, 487).

Reviewer 3 Report

The content is appropriate given that this is submitted under systematic review. The conclusion will be strengthened if a meta-analysis can be done; rather than simply summarizing the individual trial results.

Author Response

Thank you very much for your suggestion. The primary aim of this systematic review was to review systematically the latest evidence of the effects of different preoperative prehabilitation interventions (nutrition, physical activity, probiotics and symbiotics, fecal microbiota transplantation, and ghrelin receptor agonists, alone or combined) on postoperative outcomes after gastrointestinal cancer surgery. However, the results of our review are too heterogeneous in terms of study population, interventions and outcomes for a solid meta-analysis. Nevertheless, our systematic review provides perspectives for a future meta-analysis.

Reviewer 4 Report

The manuscript represents a systematic review of the literature, that was done in PRISMA. 

The introductory part could be developped. 

Some mentions about ERAS protocols are worth to be mentioned (https://www.paliatia.eu/new/2019/04/the-preoperative-management-in-palliative-colorectal-surgery-according-to-the-enhanced-recovery-after-surgery-eras-society-recommendations/). 

Also, the borderline tumors like GISTs should be mentioned. 

Author Response

The manuscript represents a systematic review of the literature, that was done in PRISMA. 

Thank you very much for reviewing the manuscript. Please find below the point-by-point detailed response to the remarks.

The introductory part could be developped. Some mentions about ERAS protocols are worth to be mentioned (https://www.paliatia.eu/new/2019/04/the-preoperative-management-in-palliative-colorectal-surgery-according-to-the-enhanced-recovery-after-surgery-eras-society-recommendations/). 

Thank you for your relevant comment. We agree that ERAS protocols should be mentioned in the introduction and not only in the discussion. We have now referred to the ERAS guidelines in the introduction (line 79): “Despite promising results, the latest Enhanced Recovery After Surgery (ERAS) guidelines for perioperative care in elective colorectal surgery and esophagectomy has mentioned a low level of evidence of prehabilitation programs [16, 17]. Indeed, the efficiency of these interventions has been debated in previous systematic reviews in GI cancers and standardization in terms of clinical pathway is needed [18-20].”

Also, the borderline tumors like GISTs should be mentioned. 

The aim of the manuscript was to review systematically the latest evidence of the effects of different preoperative prehabilitation interventions (nutrition, physical activity, probiotics and symbiotics, fecal microbiota transplantation, and ghrelin receptor agonists, alone or combined) on postoperative outcomes after GI cancer surgery. Thus, we included all type of GI cancers in our search strategy (filter including: “Gastrointestinal" OR "Digestive"). In response to the suggestion of the reviewer, we added “GIST” and “Gastrointestinal Stromal Tumor” to our previous filter, but we did not find additional articles.